# Portable Technology to Measure and Visualize Body-Supporting Force Vector Fields in Everyday Environments

**DOI:** 10.3390/s25133961

**Published:** 2025-06-25

**Authors:** Ayano Nomura, Yoshifumi Nishida

**Affiliations:** Department of Mechanical Engineering, Institute of Science Tokyo, Tokyo 152-0033, Japan; nishida.y.af@m.titech.ac.jp

**Keywords:** product safety for older adults, life-centric design, data visualization, sensing technology

## Abstract

Object-related accidents among older adults often result from inadequately designed furniture and fixtures that do not accommodate age-related changes. However, technologies for quantitatively capturing how furniture and fixtures assist the body in daily life remain limited. This study addresses this gap by introducing a portable, non-disruptive system that measures and visualizes how humans interact with environmental objects, particularly during transitional movements such as standing, turning, or reaching. The system integrates wearable force sensors, motion capture gloves, RGB-D cameras, and LiDAR-based environmental scanning to generate spatial maps of body-applied forces, overlaid onto point cloud representations of actual living environments. Through home-based experiments involving 13 older adults aged 69–86 across nine households, the system effectively identified object-specific support interactions with specific furniture (e.g., doorframes, shelves) and enabled a three-dimensional comparative analysis across different spaces, including living rooms, entryways, and bedrooms. The visualization captured essential spatial features—such as contact height and positional context—without altering the existing environment. This study presents a novel methodology for evaluating life environments from a life-centric perspective and offers insights for the inclusive design of everyday objects and spaces to support safe and independent aging in place.

## 1. Introduction

In the centenarian era, society should be redesigned so that older people can live safely and independently, despite functional abilities declining with age. The changes in physical and cognitive abilities that accompany aging increase the risk of falls [1]. For this reason, the causes of falls have been reconsidered as problems of poor design, i.e., designs that do not take into account functional changes in the aged. Hence, for older adults, whose life functions continue to change, a new design methodology is required. Figure 1 summarizes the study area and what was hoped to be achieved.

Traditional fall prevention strategies often involve physical modifications such as installing handrails or ramps [2]. However, these methods are typically only implemented when a certain change in physical function occurs, such as when climbing up and down steps becomes difficult; thus, support for minor functional changes before there is a noticeable decline remains an issue. While theoretical frameworks like the social determinants of health (SDHs) have highlighted the importance of environmental factors [3], qualitative investigations into specific environmental factors in households have also been conducted [4]. However, methods to quantitatively assess these factors remain limited.

Based on accident [5] and video data [6], we previously found that older adults often engaged in body-supporting behaviors—such as using their hands to stand up or climb steps—through interactions with everyday objects. When these objects lack sufficient body-supporting properties, the risk of falls and object damage increases. This aligns with previous qualitative studies [7] that showed that small mismatches between daily actions and the environment can lead to balance loss. Therefore, body-supporting properties are critical environmental factors in fall prevention and maintaining independence in older adults.

Techniques for measuring object use in everyday spaces include embedding sensors into the environment, such as floor sensors [8], bed sensors [9], handrail sensors [10], and toilet sensors [11]. Muheidat et al. used carpet sensors to detect anomalies [8], while Gaddam et al. employed pressure sensors to monitor unusual patterns [9]. Hamada et al. developed handrail sensors to observe changes in physical function [10], and Molenbroek et al. created a toilet prototype for older adults and disabled users to collect real-world usage data [11]. Although these embedded sensors offer valuable insights, they require specialized installation and significant environmental modifications. Therefore, they are unsuitable for evaluating the body-supporting properties of arbitrary objects in diverse everyday environments. Consequently, our understanding of such properties remains limited.

Recent advances in sensing technology have enabled behavior analyses not only in specific locations but across entire living environments. Wearable force sensors —such as those mounted on shoes and hands—can now measure support forces in daily settings with minimal burden [12,13,14,15]. In addition, systems like Microsoft’s Azure Kinect provide simultaneous skeletal tracking and 3D environmental data [16,17]. Progress in virtual reality and motion capture technologies for the hands and fingers are contributing to the growth in this field [18]. These advancements have enabled the analysis of subtle daily behaviors that were previously difficult to measure. By mapping body-supporting behaviors across entire living environments, we can translate the findings from epidemiological and quantitative studies into practical insights that are applicable to everyday life for older adults.

In this paper, we propose a whole-space body-supporting force field measurement and visualization system for elucidating the body-supporting properties of everyday spaces, based on the view that the insufficient “body-supporting properties” of everyday objects and environments are key factors affecting fall risk and independent living. By combining wearable sensors and markerless skeletal recognition, we visualize the body-supporting force field within a reconstructed 3D model of the living environment without embedding sensors into the environment. We also evaluate the system’s usefulness for evaluating body-supporting behavior of individualss and the body-supporting properties of everyday objects through in-home experiments with older adults. Building on this, we aim to address the following research question: 

**Research Question:** Can we develop a method to analyze and evaluate the extent to which the current home environment supports the physical functions of older adults, by using wearable sensors and markerless skeletal tracking?

**Hypothesis:** We hypothesize that a combination of wearable force sensing and markerless skeletal tracking enables the effective evaluation of body-supporting behaviors and environmental support characteristics in the real living spaces of older adults.

This paper is structured as follows: Section 2 proposes a whole-space body-supporting force field measurement and visualization technology. Section 3 describes the experimental conditions under which the system proposed in Section 2 is validated. Section 4 presents the results obtained from the in-home evaluations. Section 5 discusses the implications of the findings in the context of environmental design and user behavior. Finally, Section 6 offers conclusions drawn from the findings and outlines potential avenues for future research.

## 2. Measurement System and Components

In this study, we developed a portable body-supporting force vector field visualization system that measures the magnitude, direction, and position of the force applied during a body-supporting action to measure the body-supporting properties of the objects in everyday spaces. Figure 2 shows the structure of the proposed system, which consists of four sensors: two glove sensors, an iPad, and an RGB-D camera. One glove sensor is a wearable force sensor and the other is a hand motion capture sensor. The iPad utilizes a sensor for scanning the daily environment. To reproduce the everyday space on a computer and capture the body-supporting position, we acquire the shape of the environment as a point cloud to reproduce it on a computer. The skeletal recognition technology of the RGB-D camera is used as a motion tracking component, thereby enabling the portability of the entire system. The four sensors are described in Section 2.1.

Posture, hand posture, and force data are acquired by moving the three sensors simultaneously. The method for integrating the data from the four sensors is described in Section 2.2.1, whereas the implementation of the visualization system using the integrated data is described in Section 2.2.2.

### 2.1. Sensors

We employed a motion capture glove and force sensors to measure the force and how the hand is used when older adults support their body using their hand. An analysis of video data [6] revealed that body-supporting movements used all parts of the hand, including not only the palm but also the fingers. Therefore, in our analysis, we used FingerTPS (Pressure Profile Systems, Los Angeles, CA, USA), which can measure the force applied to the hand at multiple locations, including the fingers and palm. We also used Prime 3 Haptics (Manus, Geldrop, The Netherlands) to measure hand posture, given the relatively light weight of the device and the fact that it is worn in combination with the force sensor. The specifications of FingerTPS and Prime 3 Haptic XR can be found in Table 1 and Table 2. The MoCap glove had six IMU sensors with 9 degrees of freedom (3-axis acceleration, 3-axis gyro, 3-axis compass) per glove (one hand) and five flexible sensors that detected the degree of bending with 2 degrees of freedom, one on each finger. The hand posture could be estimated from these sensor values.

The device connecting the measurement units fixed to both wrists was wired to a child device that was connected to a PC via Bluetooth.

Figure 3 shows the relationship between the MoCap glove and force sensors, as well as how they were layered on the hand. Due to the limitation of only having six available force measurement units per hand, the force sensors were attached to four fingers (excluding the little finger), the base of the thumb, and the base of the ring finger on both hands. Sensor placement was determined based on preliminary trials, in which one of the authors performed various body-supporting behaviors such as leaning or gripping. From these observations, the thenar region, central palm, and pads of the index to ring fingers were identified as consistently active contact points and thus prioritized. The MoCap glove was used to obtain tilt angles corresponding to each force sensor. These angles were not calculated directly from the sensor’s physical coordinates but instead approximated using the orientation of the joint most representative of each sensor’s location. For the palm sensors, a shared palm joint was used to estimate the tilt of both sensors.

The proposed system acquired shape data for the everyday environment as point cloud data and visualized the body-supporting force field on the point cloud data to visualize the shape and location of everyday objects that generate body-supporting force. An iPad Pro with a LiDAR sensor was used to scan the shape data, as we wanted to keep the entire measurement system relatively simple. The software used was Scaniverse (Niantic Spatial Inc., San Francisco, CA, USA; version not recorded), a free and publicly available application. The point cloud, which was acquired with an iPad, was used to obtain the dimensions and relative locations of objects in the everyday environment.

The body-supporting position was obtained by using the hand position from the posture data obtained by the RGB-D camera equipped with skeletal recognition technology. The RGB-D camera used was a Microsoft Azure Kinect. It was possible to acquire posture data and point cloud data in the same coordinate system by using Azure Kinect, by integrating data that aligned the coordinate system of the RGB-D camera and the iPad by using point cloud data acquired from the RGB-D camera. The details of this are described in Section 2.2.1.

### 2.2. Software System

#### 2.2.1. Multi-Sensor Information Integration Technology

We integrated separately acquired time-series data from the sensors in Section 2.1 and treated the result as representing a single body-supporting behavior. The time-series data of the body-supporting behavior were obtained from three of the four sensors: the wearable force sensor, the motion capture glove, and the RGB-D camera. The measurements obtained from these three sensors were timestamped in ms and saved as files with a csv extension. For simplicity, the timestamps of all csv files were rounded to the nearest 10 ms because each sensor had a different sampling rate, all of which were below 100 Hz. This rounding approach was adopted because the sensors were operated on separate PCs, making it difficult to align the data precisely by start time. The highest sampling rate among the sensors was 90 Hz, corresponding to a time interval of approximately 11.1 ms. Therefore, rounding to 100 ms provided sufficient temporal resolution while allowing consistent timestamp alignment across all data sources. Although interpolation-based resampling to a fixed rate (e.g., 100 Hz) may reduce abrupt shifts, our method prioritized timestamp consistency and minimized the introduction of filter-related artifacts. After this process, the data were kept for times when one of the three sensors had a sampling rate; when that timestamp did not exist in the remaining two sensing datasets, linear interpolation was performed using the previous and next data points. The same time server was used for time synchronization before the start of the experiment because of the difference between the PC used for Azure Kinect and the one used for the wearable force sensor and hand motion capture device.

It was necessary to unify the coordinate systems of data acquired from the multiple coordinate systems to visualize the body-supporting force vector field. To achieve this, we unified data from three different coordinate systems: (1) the iPad Pro (Scaniverse) point cloud representing the environment, (2) the skeletal coordinates and point cloud from Azure Kinect, and (3) the hand posture from the MoCap glove.

Coordinate alignment was performed in two main steps:**Aligning iPad coordinate system to the MoCap glove**: At the beginning of the experiment, the MoCap glove was initialized in the physical space covered by the iPad scan. We assumed the initial hand orientation as a reference and computed a rotation matrix $T_{I}$ to align the iPad point cloud to that frame:$$P_{I}^{\mathrm{aligned}}=T_{I}\;\cdot \;P_{I}$$**Aligning Kinect data to the rotated iPad frame**: We applied the RANSAC-ICP algorithm [19] to align the Kinect point cloud to the already rotated iPad point cloud, yielding a transformation matrix $T_{K\to I’}$:$$P_{K}^{\mathrm{aligned}}=T_{K\to I’}\;\cdot \;P_{K},\;S_{K}^{\mathrm{aligned}}=T_{K\to I’}\;\cdot \;S_{K}$$
where $S_{K}$ denotes the skeletal coordinates from Kinect.

Although the current study did not conduct a new quantitative comparison with standard measurement tools such as pressure mats or load cells, a prior evaluation of the same visualization system using a pressure mapping sensor (SR Soft Vision, Sumitomo Riko Company Limited, Nagoya, Japan) was reported in our previous work [20]. In that study, the average localization error between the estimated contact area and the pressure mat output was 27 mm (SD: 21 mm), with a maximum error of approximately 50 mm. This resolution was sufficient to distinguish coarse contact locations (e.g., center vs. edge of a desk) in typical body-supporting scenarios.

#### 2.2.2. Visualization of the Body-Supporting Force Vector Fields

The data processed in Section 2.2.1 were used to implement the three visualization functions shown in Figure 4. The functions consisted of one main function, “visualization of the body-support force vector field”, and two built-in functions, “visualization of the gravity–horizontal component ratio” and “visualization by measurement point”. The main function displayed force vectors as arrows, whose color and length varied with the magnitude of the force. The procedure for calculating the force vectors, selecting the ones to be visualized, and determining their position was as follows: “Visualization of the gravity–horizontal component ratio” displayed the gravity and horizontal components calculated from the vectors as colors on a grayscale point cloud. This function considered whether the location had a role in supporting the feet and legs or maintaining the balance direction. The method for determining the colors on the grayscale point cloud is described later. “Visualization by measurement point” visualized the vectors shown by the main function separately for each measurement point.

First, we describe the procedure for calculating the force vectors. The force vector was calculated using the force magnitude data and the tilt angle of the hand measurement points. Let $\left[\theta_{i}\left(t\right),\phi_{i}\left(t\right),\psi_{i}\left(t\right)\right]$ be the Euler angle (roll, pitch, yaw) of the measuring point *i* of a motion at a given time *t*. The force magnitude of measurement point *i* is called $f_{i}\left(t\right)$. Assuming that the force $f_{i}\left(t\right)$ exerted by measurement point *i* at time *t* is oriented perpendicular to the measurement point, the expression for the force $f_{i}\left(t\right)$ is as in (Equation 1). Let $R_{x}\left(\zeta \right)$, $R_{y}\left(\zeta \right)$, $R_{z}\left(\zeta \right)$ be the matrices rotating ζ around the *x*, *y*, and *z* axes, respectively.(1)$$\begin{array}{c} \begin{array}{c} f_{i}\left(t\right)=R_{z}\left(\psi_{i}\left(t\right)\right)R_{y}\left(\phi_{i}\left(t\right)\right)R_{x}\left(\theta_{i}\left(t\right)\right)\left(\begin{array}{c} 0\\ 0\\ -f_{i}\left(t\right) \end{array}\right) \end{array} \end{array}$$In this equation,

$R_{x}\left(\theta_{i}\left(t\right)\right)$, $R_{y}\left(\phi_{i}\left(t\right)\right)$, and $R_{z}\left(\psi_{i}\left(t\right)\right)$ can be written as follows:(2)$$R_{x}\left(\theta_{i}\left(t\right)\right)=\left(\begin{array}{ccc} 1 & 0 & 0\\ 0 & \cos\theta_{i}\left(t\right) & -\sin\theta_{i}\left(t\right)\\ 0 & \sin\theta_{i}\left(t\right) & \cos\theta_{i}\left(t\right) \end{array}\right)$$(3)$$R_{y}\left(\phi_{i}\left(t\right)\right)=\left(\begin{array}{ccc} \cos\phi_{i}\left(t\right) & 0 & \sin\phi_{i}\left(t\right)\\ 0 & 1 & 0\\ -\sin\phi_{i}\left(t\right) & 0 & \cos\phi_{i}\left(t\right) \end{array}\right)$$(4)$$R_{z}\left(\psi_{i}\left(t\right)\right)=\left(\begin{array}{ccc} \cos\psi_{i}\left(t\right) & -\sin\psi_{i}\left(t\right) & 0\\ \sin\psi_{i}\left(t\right) & \cos\psi_{i}\left(t\right) & 0\\ 0 & 0 & 1 \end{array}\right)$$

A data point at time t is displayed as a force vector if its magnitude $f_{i}\left(t\right)$ is nonzero and its position $x_{i}\left(t\right)$ is within a certain proximity to the point cloud. The following procedure was used to select and determine the position of the force vectors to be visualized.

Calculate the position $x_{i}\left(t\right)$ of the measurement point;Find the point $p_{i}\left(t\right)$ closest to $x_{i}\left(t\right)$ in the point cloud and extract the data at time *t*, where $|p_{i}\left(t\right)-x_{i}\left(t\right)|$ is less than 10 cm;The body-supporting position of the data at time *t* extracted in step 2 is the nearest neighbor point $p_{i}\left(t\right)$ calculated in step 2;Delete data with a magnitude of $f_{i}\left(t\right)$ less than or equal to 0;Represent the force vector as an arrow at the body-supporting position in the point cloud.

The position $x_{i}\left(t\right)$ of the measurement point at time *t* was calculated by combining the hand position obtained from the whole-body posture data with the relative position of the measurement point viewed from the center of the hand obtained from the hand posture data. The representative points for the right and left hands were HAND_RIGHT and HAND_LEFT, respectively, based on the 33 skeletal points estimated by Azure Kinect. The position of the measurement point from the MoCap glove was relative to the center of the hand. Therefore, the body-supporting position in the point cloud coordinate system was the sum of the representative point of the hand and the relative position of the measurement point viewed from the center of the hand.

We implemented the “visualization of the gravity–horizontal component ratio” shown in Figure 4 using the vectors selected above. The locations where the body supporting force vector field was generated are colored red to blue according to the proportion of the gravity direction, and the locations where the vector field was not generated are displayed in grayscale. The color of the locations where vector fields are generated should be red when the ratio of the gravity direction is large (i.e., the vector direction is closer to the gravity direction) and blue when the ratio of the gravity direction is small (i.e., the vector direction is closer to the horizontal direction).

The color determination method is explained below. Let *n* be the number of points in the daily life environment point cloud data, and let P(j) be the coordinates of the *j*th (1≤j≤n) point. Let $x_{i}\left(t\right)$ be the position of the measurement point at time *t*, and let $f_{i}\left(t\right)$ be the force vector. The range within which $x_{i}\left(t\right)$ is a body-supporting force is defined as P(j)(m≤j≤l) within a radius of 3 cm from the nearest point in the point cloud P(j)(1≤j≤n), taking into account the size of the hands and fingers. The point group used in this study was oriented in the positive direction of the *z*-axis upward in the direction of gravity. Therefore, the gravity direction component of force vector $f_{i}\left(t\right)$ was the *z*-axis component of vector $f_{i}{\left(t\right)}_{z}$. The gravity component fraction of the force vector at the position $x_{i}\left(t\right)$ of the measurement point at time *t* is then given by $\frac{|f_{i}{\left(t\right)}_{z}|}{|f_{i}\left(t\right)|}$. If $t_{S}<t<t_{E}$ is the time when body retention occurs in P(j), the gravity component fraction g(P(j)) of P(j) is calculated as in (Equation 5). There were 12 measurement points, so *i*(1≤i≤12).(5)$$\begin{array}{c} g\left(P\left(j\right)\right)=\frac{\sum_{t=t_{S}}^{t_{E}}\sum_{i=1}^{12}\frac{|f_{i}\left(t\right)z|}{\left|fi(t)\right|}}{t_{E}-t_{S}} \end{array}$$

## 3. Evaluation in Laboratory and Everyday Environments

### 3.1. Laboratory Verification of the Proposed System

We validated the proposed system in three aspects. The first was to verify the accuracy of the motion capture glove’s orientation. The second was to evaluate the accuracy of the body-supporting position displayed by the proposed system. The third was to verify whether it was possible to visualize the body-supporting force vector field using the proposed system in a laboratory simulation of a living environment.

First, we discuss verifying the accuracy of the motion capture glove’s orientation, which determines the direction of the body-supporting force vector in the proposed system. As described in Section 2.1, the orientation of the palm and fingers, which were equipped with force sensors, was obtained from the orientation of a virtual hand that was moved based on the data acquired by the motion capture glove. For accuracy verification, we measured the deviation between the actual and obtained orientation values when the palm was moved in the specified directions for roll, pitch, and yaw.

Figure 5 shows how the measurements were taken by moving the object in the roll, pitch, and yaw directions, each based on a protractor. The figure illustrates that measurements were taken twice for each direction at $30^{\circ }$ intervals, within a range from $0^{\circ }$ to $180^{\circ }$.

The catalog specifications of the finger sensor (FingerTPS) indicate a valid measurement range starting from approximately 44.6 N, as summarized in Table 1. However, in the present study, the forces generated during body-supporting movements were expected to be much smaller. Therefore, we conducted a calibration experiment to evaluate whether the sensor could maintain sufficient accuracy within a lower force range. Specifically, we used a reference force gauge (FGP-10 (Nidec Drive Technology Corporation, Kyoto, Japan)) to apply forces ranging from approximately 0 to 23 N. To simulate actual usage conditions, we repeatedly applied and released forces by manually pressing the sensor with varying intensities. The finger sensor readings were baseline-corrected by subtracting the initial offset under unloaded conditions. Figure 6 shows the experimental setup.

To validate whether the body-supporting force vector field could be visualized using the proposed system, we conducted experiments in a laboratory simulating a living environment. One male participant in his twenties performed body-supporting movements. The participant performed actions such as leaning his hands against a wall while putting on and taking off shoes and placing his hands on a table while sitting down and standing up from a chair.

In this study, we did not conduct a separate validation of the positional accuracy of the visualized body-supporting force field. However, in our previous study [20], a similar verification was performed using the same pressure distribution sensor (SR Soft Vision, Sumitomo Riko, resolution 22 mm). The results showed a mean error of 27 mm, a standard deviation of 21 mm, and a maximum error of approximately 50 mm, which were considered sufficient to identify general contact areas such as the edge of a desk or the armrest of a chair. Based on these findings, additional accuracy validation was omitted in the present study.

### 3.2. Validation of Understanding Elderly People’s Body-Supporting Behaviors in an Everyday Environment

We conducted experiments to measure the body-supporting movements of older adults in their daily environments to validate whether the system described in Section 2 was useful for understanding and deepening our knowledge of body-supporting movements. The participants were 13 older adults (5 men, 8 women; age range, 69–86 years), and the experiment was conducted in nine houses where the participants lived. All participants were recruited in cooperation with the Tokyo Metropolitan Institute for Geriatrics and Gerontology.

The experimental procedure was as follows. First, an interview survey was conducted aiming to assess the participants’ usual body-supporting movements. Then, based on the results of the interview survey, we decided on an experimental site and recorded the participants performing their usual body-supporting movements at that location. This experiment was conducted with the approval of the Human Subjects Research Ethics Review Committee of the Tokyo Institute of Technology (Permission No. 2023279). Written informed consent was obtained from all participants before the study began.

Figure 7 shows the tools used in the experiment. The figure shows that all the necessary equipment for conducting the experiments could be packed into three bags, which highlights the portability of the proposed system.

## 4. Results

### 4.1. Verification Results

#### 4.1.1. Orientation Accuracy and Its Implication

The results of the accuracy validation of the rotation angle of the motion capture glove are shown. Using a protractor, measurements were taken at seven points each for roll, pitch, and yaw, within a range from $0^{\circ }$ to $180^{\circ }$ and with a resolution of $30^{\circ }$. For the yaw angle, since a reference point could not be identified, the average error was calculated based on the differences from the $0^{\circ }$ measurement point at six points. The results showed that the average errors for roll, pitch, and yaw were $4.9\pm 2.3^{\circ }$, $2.3\pm 1.1^{\circ }$, and $3.0\pm 2.2^{\circ }$, respectively, indicating that the system provided sufficient accuracy for determining the application of force and calculating vector direction.

#### 4.1.2. Force Sensor Calibration and Reliability

In addition to orientation accuracy, we also evaluated the reliability of the finger sensor in detecting supporting force. Figure 8a presents the time-series comparison of the finger sensor and the force gauge. The two signals showed consistent trends, confirming the sensor’s capability to detect changes in applied force. Figure 8b shows the regression analysis of the finger sensor and force gauge outputs. The regression equation was y=0.84x+1.96, with $R^{2}=0.85$ and RMSE = 1.61 N. These results indicate that while the finger sensor underestimated the force slightly and had a nonzero offset, its output remained sufficiently reliable for a qualitative analysis of supporting force. These results demonstrate that despite the manufacturer’s specified lower limit of 44.6 N (Table 1), the sensor performs reliably within the 0–23 N range observed in everyday body-supporting behavior.

#### 4.1.3. Force Vector Field Visualization

The results for the verification of visualizing the body-supporting force vector field using the proposed system in a laboratory-simulated living environment are presented in Figure 9, which shows the data obtained from the verification experiment visualized as a body-supporting force vector field using the proposed system, along with the image data captured at the time of the experiment. Our results confirmed that the body-supporting force vectors could be displayed for the actions of “holding the wall while putting on and taking off shoes” and “holding the desk when sitting down and standing up from a chair” that were observed in the verification experiment.

### 4.2. Results of the Home-Visit Investigation

It is well recognized that older adults often require hand support when performing daily activities such as dressing, standing up, or getting out of bed [5,7]. However, specific evidence on which everyday objects and environmental features are actually used for body support in these moments has been limited. Based on the in-home observational study method described in Section 3.2, this study clarified which specific everyday objects and environmental elements were used to support the body during activities that require physical support, such as sitting down and standing up, lying down and getting up, putting on and taking off clothes and shoes, and walking up and down stairs. A wide variety of household items were observed to be used for body support, including tables, low tables, chairs, footrests, entryway shelves, handrails, kitchen counters, TV stands, beds, windows, ceilings, and doors. The visualization results of the body-supporting forces during these behaviors are shown in Figure 10.

## 5. Discussion

### 5.1. System Capabilities

In this subsection, we explore how the three built-in visualization functions of the proposed system—palm-only force vectors, finger-only force vectors, and gravity–horizontal component ratios—can be used to capture subtle differences in how individuals apply body-supporting forces in real-life situations. We focus on a bedroom case study to examine the potential of region-specific sensing and visualization in understanding user behavior during transitional postures.

Figure 11 illustrates the body-supporting force field of a couple sleeping on a futon (traditional Japanese bedding consisting of a thin mattress and blanket) placed on tatami mats. The upper section (**a**) corresponds to the man, who is 168 cm tall and sleeps on the left side, while the lower section (**b**) depicts the woman, who is 153 cm tall and sleeps on the right.

In (**a**), the man is observed to support his body by grasping the corner of the shelf along the left wall, primarily using the base of his thumb and his index finger. The visualized vector fields suggest different usage patterns depending on his posture: when lying down, he appears to hook his index finger on the upper edge of the shelf, whereas during the process of standing up, he shifts to placing pressure on the base of his thumb. These interactions exert multi-directional forces on the shelf corner.

In contrast, (**b**) shows the woman grasping the nearby sliding door with her thumb and middle finger. Given the absence of alternative support structures on her side, it is inferred that the sliding door’s accessible height and thickness made it a practical choice for body support during rising motions.

Both (**a**) and (**b**) demonstrate the use of body-supporting force fields in locations that are not inherently designed for postural assistance. The shelf in (**a**) is situated at a height of approximately 110 cm, making it less accessible from a lying position. Meanwhile, the door in (**b**), while conveniently placed at a 50–60 cm height, is not a fixed support but a movable component. These observations underscore the adaptive behavior of individuals in utilizing nearby objects for postural support, regardless of the object’s intended function or stability.

This case study suggests that separating sensor data by hand region can help reveal subtle variations in how users engage with support structures depending on the phase of movement and object height. Although further validation is needed, such detailed visualization capabilities could be leveraged in future work for ergonomic evaluation or personalized environmental design.

### 5.2. Case-Based Observation and Interpretation

This section presents a case-based interpretation of the results obtained using the proposed system as part of our approach to examine the hypothesis introduced in Section 1. By analyzing representative examples of daily behaviors in actual living environments, we aim to explore how wearable force sensing and markerless skeletal tracking can reveal the ways in which older adults interact with and utilize their surroundings for postural support. This discussion focuses on the contextual variations in support usage, the adaptive strategies observed, and the practical implications for designing environments suitable for postural support. Taken together, these observations support our original hypothesis that combining wearable force sensing with markerless skeletal tracking enables effective, context-aware evaluation of body-supporting behaviors in real-life environments.

#### 5.2.1. Even Non-Assistive Elements Are Widely Used for Body Support

The overall results indicated that when older adults perform movements requiring postural support—such as standing up, sitting down, or changing shoes—they tend to use whatever objects are within reach, regardless of whether those objects were originally designed to provide support. These findings align with the concept of environmental affordances [21], suggesting that users tend to creatively repurpose familiar objects in their environment to meet their physical needs. In this subsection, we focus on cases where users creatively utilized everyday items that were not intended as assistive devices. These observations illustrate how the proposed system can reveal such behavior and affordance-driven interactions in real-world living environments, contributing to a more nuanced understanding of bodily support usage beyond formal assistive tools.

Figure 12 shows a visualization of results for the entrances to a detached house and an apartment. In Japan, most detached house entrances have a step that is about 30 cm higher than the entrance and most apartment entrances are very narrow, with widths about the same size as a human wingspan. Previous studies have discussed the risk of tripping on steps in entryways and the difficulty of perceiving elevation changes in narrow spaces [4,5,7]. However, in this study, we observed users grasping the top of a shoe cabinet during transitional movements such as putting on or taking off shoes, suggesting that the cabinet may play an important role in balance recovery in these constrained environments. The left side of Figure 12 shows the force fields for a shelf and steps. In an entryway with a relatively large step, users sit on the step, where they can put on footwear while supporting their body. The right side of Figure 12 shows the force fields for the shelf and side walls. The lack of a step to sit on suggests that the individuals using this space may stabilize their posture by placing their hands on either side wall when standing up and putting on footwear.

Figure 13 shows a visualization of the force field for the shelf in front of a handrail at an entrance. The handrail was installed to assist adults in changing shoes at the entrance or going up the step; however, the results showed that the shelf in front of the handrail was used for support rather than the handrail itself. This finding suggests that it is possible to assist movement by installing a shelf for support without the need for installing a handrail on the wall. Similar findings have been reported in previous studies, which indicate that assistive features such as handrails are not always utilized even when they are available [22]. The present result aligns with these observations, providing empirical evidence that real-life use predominantly depends on the interaction between the user and the spatial context—not just the presence of a support structure. While many studies have experimentally evaluated the balance recovery characteristics of handrails [23,24,25], these findings highlight that, in real-life environments, such handrails may not be used as expected.

These results (i.e., those shown in Figure 12 and Figure 13) suggest that the usability of support elements is not determined solely by their intended function or presence but also by their accessibility, perceived stability, and spatial positioning. In this case, the shelf afforded a more natural grip location than the wall-mounted handrail, highlighting the importance of contextual affordance in real-world environments.

Another illustrative example of creative support behavior is seen in the use of a sliding door during rising movements from a lying position on a futon. As shown in Figure 11, the woman on the right side, lacking alternative support structures, grasped the sliding door with her thumb and middle finger to aid in posture stabilization. Although the door was not designed to serve as a supportive structure—and indeed is a movable architectural element—its accessible height and thickness made it a practical choice in this context. This observation further supports the idea that older adults flexibly repurpose available features in their environment for postural support, depending not only on function or safety but also on location, accessibility, and habitual familiarity.

#### 5.2.2. Contextual Factors Determine Support Strategies Even with Identical Furniture

In the previous section, we highlighted how users repurposed a wide range of everyday objects—many not originally intended for postural assistance—for body support. Here, we extend the discussion by focusing on another important dimension: how even identical furniture can afford different support strategies depending on the spatial context, user intent, and physical movement. By analyzing body-supporting force fields from two homes where the same type of furniture (zaisu chairs) was used, we explore how contextual and task-based variations affect the nature and direction of applied force. This offers a deeper understanding of the interaction between the environment, posture, and object affordance in daily activities.

The visualization, including the two built-in functions, allows for the differences in gripping due to shape and positioning to be considered. Figure 14 shows two ways of using zaisu chairs in two different houses. The images show visualizations of the body-supporting force fields when the subject uses a low table and a zaisu chair. The pictures at the bottom of the figure show visualizations of a zaisu chair in front of a Buddhist altar. Both the upper and lower visualizations use two built-in functions. From left to right, the figure shows the visualization of the force field only at measurement points on the palm, visualization of the force field only at the finger measurement points, and visualization of the gravity–horizontal ratio. The images in the top panel of the figure show that the subject is supporting themselves with the palm more than with the fingers and using the elbow rest of the zaisu chair in the gravity direction. On the other hand, the images at the bottom of the figure show that the subject is holding on with the fingers more than with the palms and supporting her body by holding the front side of the elbow rest of the zaisu chair in the horizontal direction. The environment shown in the upper part of the figure is more restricted to forward movement than the environment in the lower part because of the table in front of the chair; thus, it can be considered that the force required to stand up vertically and push downward by gravity has increased. Because a Buddhist altar is in front of the subject in the lower figure, it is suggested that the subject is completing a task such as offering incense and wants to stand up in a forward direction, which may increase the forward load. Thus, even if the object is the same chair, it can be used in different ways depending on the height of the elbow rest, user, placement, and intended use.

These results demonstrate that even identical pieces of furniture can afford different modes of support depending on the spatial context, functional intent, and user’s behavior. The contrast between the vertical and horizontal force directions suggests that users adapt their support strategies based on the available space and task orientation. This highlights the importance of evaluating not only the design of individual furniture items but also their placement and usage scenarios in the home. From a design perspective, incorporating variable grip surfaces or affordance-enhancing features—such as textured edges or angled armrests—may support intuitive postural adjustments in varied everyday situations.

#### 5.2.3. Small Forces Can Play a Crucial Role in Postural Stability

The proposed system, which can specifically measure body-supporting force, has enabled us, for the first time, to determine the magnitude of the force used on everyday objects for postural support.

Figure 15 presents the time-series graphs of the body-supporting force magnitude, with the vertical axis indicating force (N) and the horizontal axis showing time (s). As shown in Figure 15, this study revealed that participants applied small forces—typically around 4 N—to nearby items such as TV racks, doors, and walls while putting on or taking off footwear or clothes. Absolute, horizontal, and parallel components are plotted. The two blue-highlighted force fields represent instances in which participants lightly touched TV racks, walls, or doors with measured horizontal support forces of approximately 3.9 N for TV racks and 2.7 N for doors or walls. Notably, these values were derived from the sensor outputs shown in Figure 15 and their accuracy was validated by the verification experiments presented in Section 4.1.2.

Although previous studies have examined the body-supporting forces applied to handrails and desks using controlled experiments [25,26], little is known about the actual magnitude of forces exerted on general household items. While the concept of “light touch” has been discussed in the context of postural control [27,28], this study provides the first empirical evidence that such small forces are actually utilized in daily life for body support using familiar household items.

In summary, the current findings underscore the adaptive use of everyday elements for postural support. While individual environments are fixed, analyzing behavior across multiple homes opens new possibilities for the generalizability of the findings and enhancing support strategies.

## 6. Conclusions

A lack of body-supporting objects in daily environments is a key factor affecting the independence of older adults’ and their risk of falls. To evaluate the body-supporting properties of various environments, we proposed a system that measured and visualized body-supporting force vector fields throughout living spaces. To assess the system’s practical applicability, we conducted in-home evaluations with 13 older adults between the ages of 69 and 86 across nine different residences. The findings from this study are summarized below.

**Widespread Use of Everyday Items:** A wide range of household items—including tables, chairs, shelves, doors, walls, and window frames—were used for body support during everyday movements such as standing up, sitting down, dressing, or undressing, regardless of whether they were originally designed as assistive tools.**Environmental Context Influences Support Strategies:** For instance, in entryways where a shelf was placed in front of a wall-mounted handrail, participants were observed using the shelf rather than the handrail to support themselves when putting on or taking off shoes. Results such as this highlight how spatial layout and accessibility shape support behavior.**Variability in Grip Strategy and Force Direction:** Even for the same object—such as the armrest of a chair—the hand placement and direction of force varied depending on the user, action, and environmental context. This underscores the diversity of postural support strategies employed in real-life settings.**Significance of Small Forces:** The measured horizontal support forces were typically small, averaging around 4 N, yet they played a crucial role in maintaining balance. This provided quantitative evidence of the importance of “light touch” in everyday activities.**Implications for Environmental Design:** The measured data obtained by the proposed system can inform the reassessment of existing design elements and contribute to the development of intuitive support tools that reflect actual usage patterns.

Overall, this study demonstrated that body-supporting force vector fields could be effectively measured and visualized in daily environments without embedding sensors into the surroundings. This approach provides a new foundation for evaluating postural support in real-world contexts, with implications for enhancing independence and fall prevention in aging populations.

### Limitations

The challenges and future prospects of this research are summarized as follows.

The glove-based sensor system may influence natural movement patterns, as participants are required to wear it during measurement. While the current system enables the identification of variables useful for evaluating postural support, future work may benefit from integrating more passive or embedded sensing techniques.This study focused on force vector fields without assessing participant posture. Future research should investigate the postural configurations associated with balance loss and classify support adequacy based on these combined factors.The current study did not evaluate whether the observed support strategies were appropriate or ideal. However, the findings suggest that everyday items may provide sufficient support without requiring structural modifications, such as installing handrails. Designing everyday objects/furniture with inherent supportive properties is a promising direction.As more behavioral data are accumulated, there is potential for predictive modeling of user behavior and proactive environmental design. Future studies could explore how body-support capabilities change with environmental alterations, which could contribute to data-driven, user-centered design strategies.The sample size of this study was relatively small, involving only 13 participants across nine households. This number was determined based on the practical constraints of deploying sensor systems in real-world living environments and aligns with prior exploratory research on in-home assistive evaluations. While sufficient for a proof-of-concept demonstration, larger-scale studies will be required to generalize the findings, conduct statistical analyses, and capture variability across diverse living contexts.

## Figures and Tables

**Figure 1 sensors-25-03961-f001:**
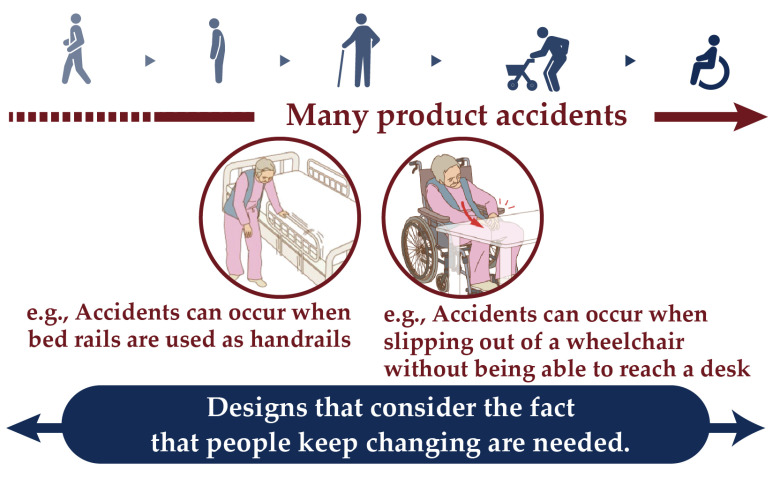
The need for designs that consider changes in life functions.

**Figure 2 sensors-25-03961-f002:**
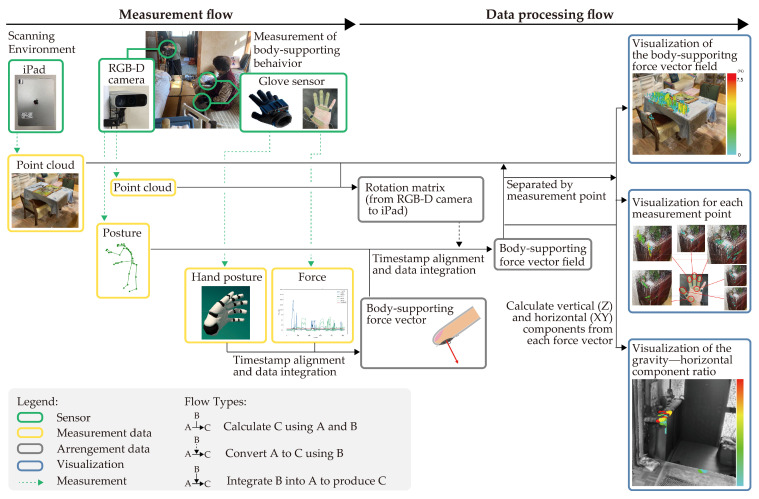
Structure of the body-supporting force vector field visualization system.

**Figure 3 sensors-25-03961-f003:**
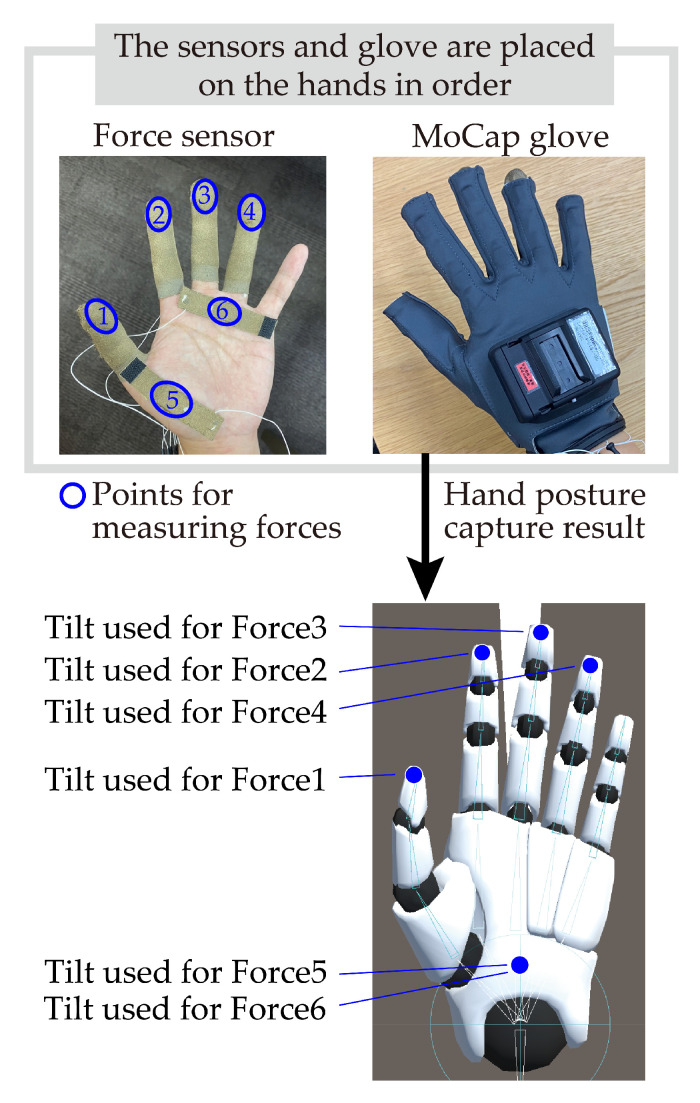
Relationship between MoCap glove and force sensors.

**Figure 4 sensors-25-03961-f004:**
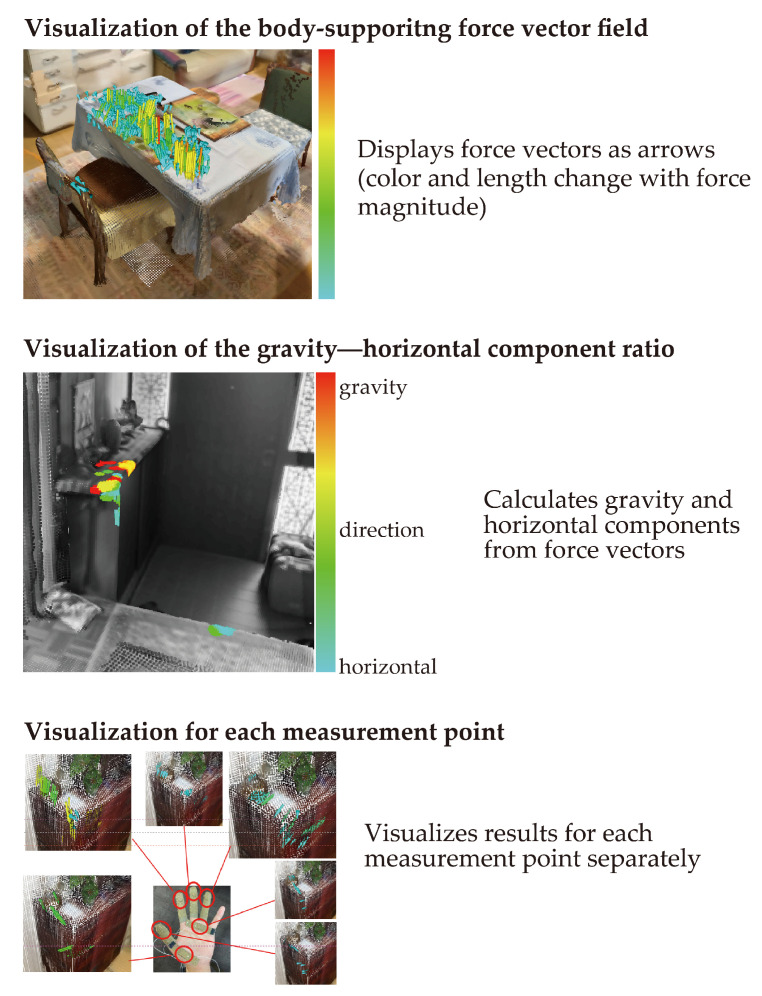
Visualization of the body-supporting force vector field and two built-in functions.

**Figure 5 sensors-25-03961-f005:**
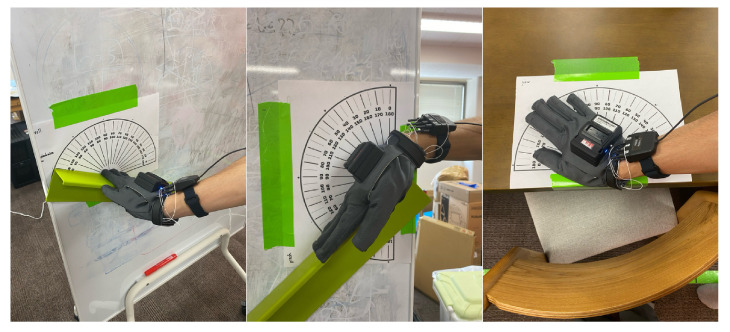
Verifying the accuracy of the motion capture glove’s orientation.

**Figure 6 sensors-25-03961-f006:**
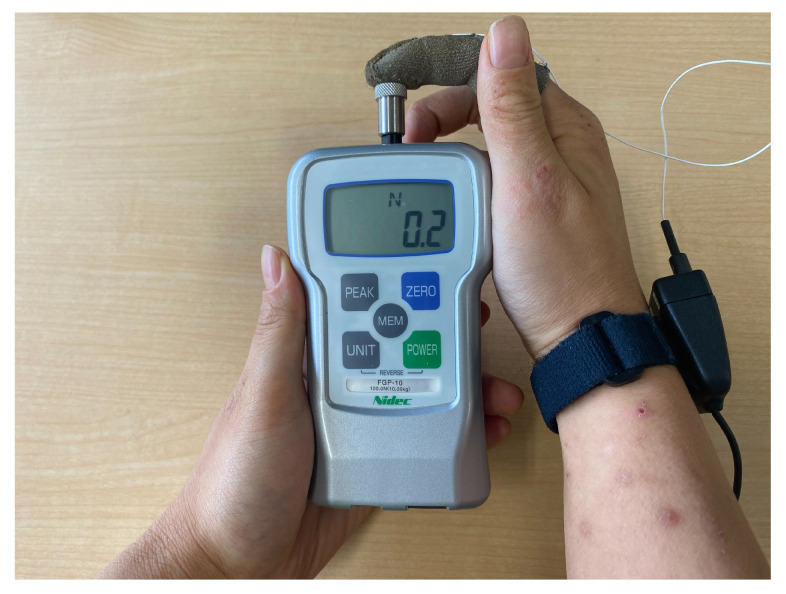
Experimental setup for sensor calibration using a force gauge and the finger-mounted sensor.

**Figure 7 sensors-25-03961-f007:**
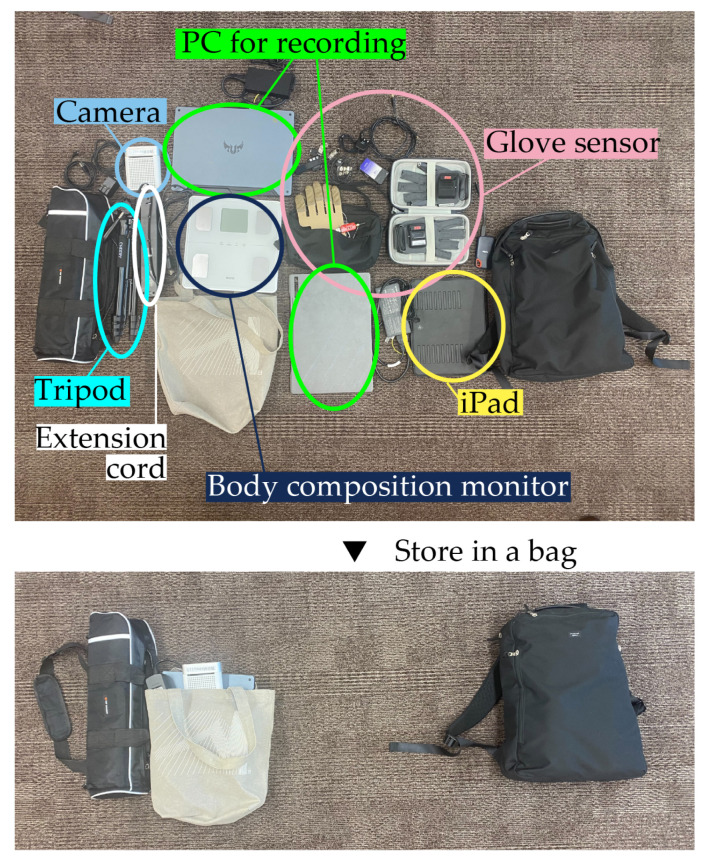
The tools needed for the experiment.

**Figure 8 sensors-25-03961-f008:**
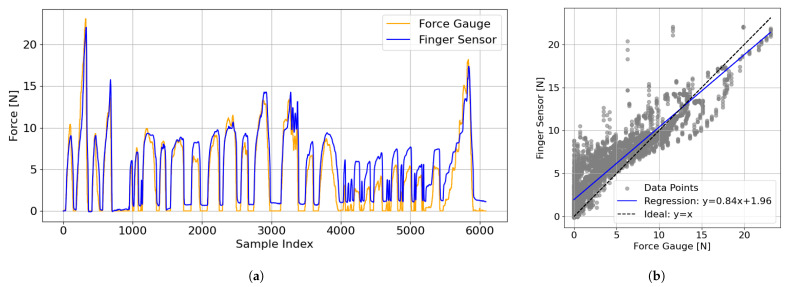
Comparison between the finger sensor and the force gauge. (**a**) Time-series comparison of the finger sensor and the force gauge. (**b**) Regression analysis involving the finger sensor and the force gauge.

**Figure 9 sensors-25-03961-f009:**
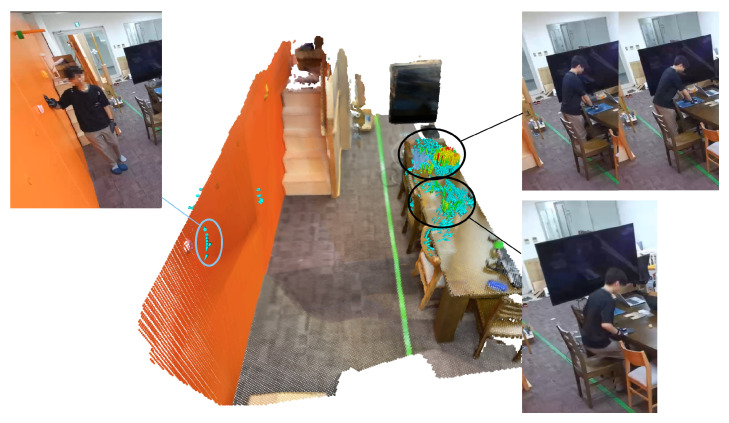
The results of the laboratory verification.

**Figure 10 sensors-25-03961-f010:**
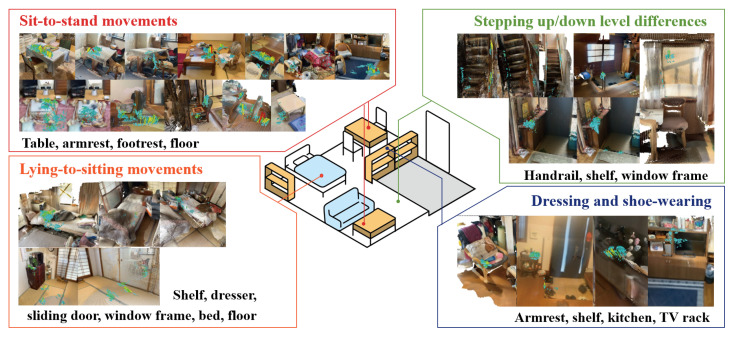
Visualization of all the body-supporting force fields obtained in the study.

**Figure 11 sensors-25-03961-f011:**
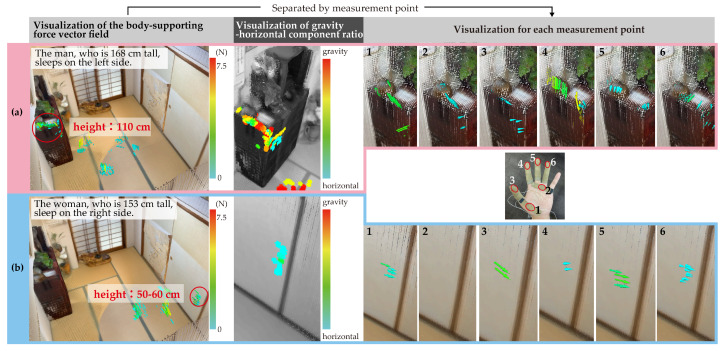
Visualization of the force field of a couple sleeping on a futon (Japanese bedding similar to a thin mattress and blanket) atop tatami mats. (**a**) shows the man (168 cm tall) sleeping on the left side, and (**b**) shows the woman (153 cm tall) sleeping on the right side. From left to right, the figure presents (i) visualization of the body-supporting force vector field, (ii) visualization of the gravity–horizontal component ratio, and (iii) visualization of the force field for each measurement point. In section (iii), the central image of the hand shows measurement point numbers 1–6, which correspond to specific locations from the fingertip to the palm. The visualizations displayed directly above and below this image illustrate the force vector fields measured at each of these respective points.

**Figure 12 sensors-25-03961-f012:**
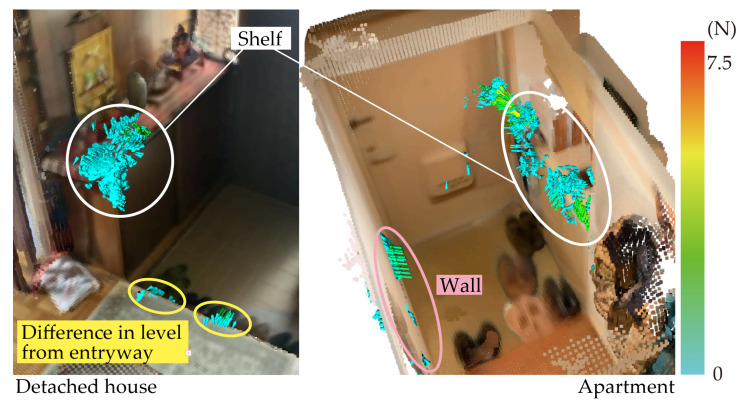
Comparison of detached house and apartment entrances.

**Figure 13 sensors-25-03961-f013:**
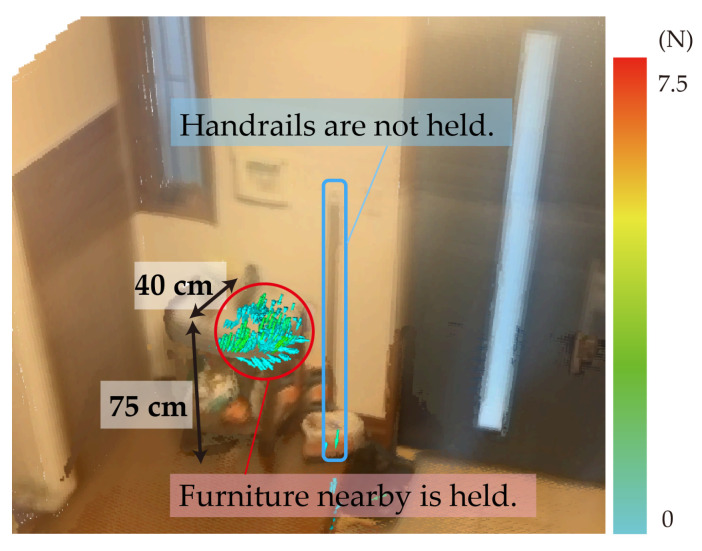
The shelf in front of the handrail on the wall is used to support the body at the entrance.

**Figure 14 sensors-25-03961-f014:**
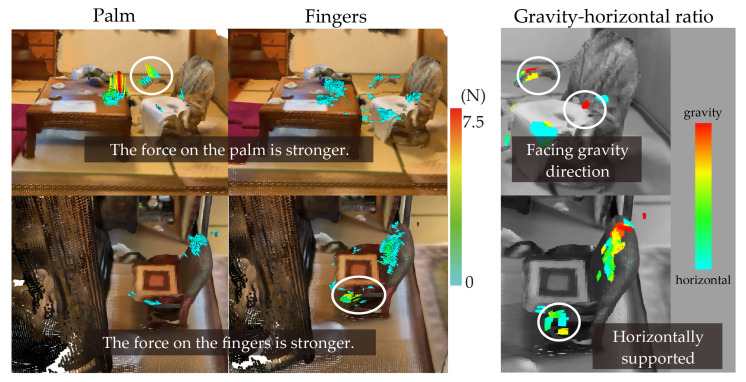
Visualization of two ways of using zaisu chairs for support. White circles indicate representative contact areas used for support in each condition.

**Figure 15 sensors-25-03961-f015:**
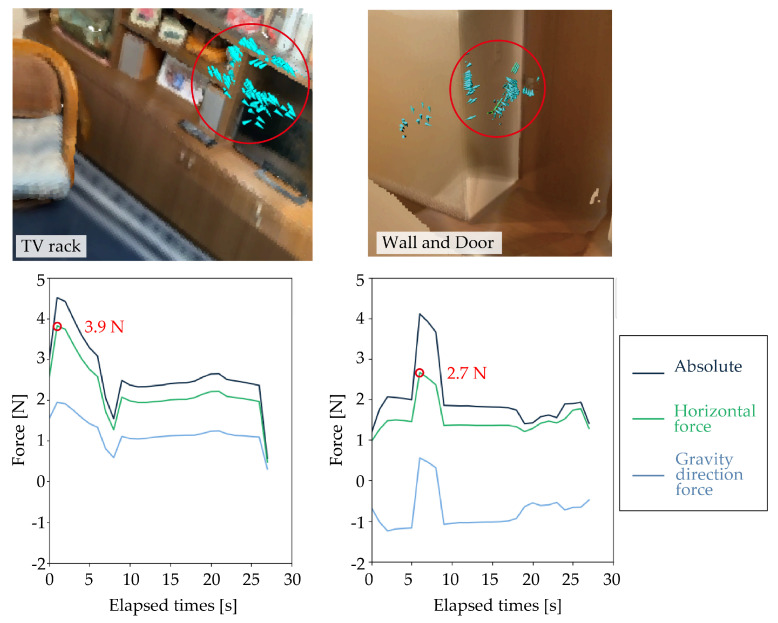
Comparison of support forces during light contact with a TV rack and wall/door. Red circles in the graphs indicate peak force values measured at the fingertip.

**Table 1 sensors-25-03961-t001:** Specifications of FingerTPS.

Parameter	Value
Thickness	2 mm to 3 mm
Measuring range	44.6 N to 222.8 N
Sensitivity	0.44 N
Temperature range	0–50
Sampling rate	40 Hz
Operating time	max 4 h

**Table 2 sensors-25-03961-t002:** Specifications of Prime 3 Haptic XR.

Parameter	Value
Latency (communication delay time)	≦75 ms
Sampling rate	90 Hz
Battery life	10 h
Charging time	3 h
Weight	134 g
Finger sensor type	2-DOF flexible sensor × 5
	9-DOF IMU sensor × 6
Finger flexible sensor durability	1 million bending cycles
Glove material	Polyester 77%, Spandex 23%

## Data Availability

The data presented in this study are available upon request from the corresponding author due to legal or ethical reasons.

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
