# Peer review of "Portable Technology to Measure and Visualize Body-Supporting Force Vector Fields in Everyday Environments"

_sensors, 2025, doi:10.3390/s25133961_

Round 1
Reviewer 1 Report
Comments and Suggestions for Authors
This paper proposes a portable system that includes wearable sensors (e.g., force sensors, motion capture gloves), RGB-D, and LiDAR-based enviroment scanning to visualize the body-supporting behavior in different environmental settings. The work is meaningful to understand the preferred furniture that users to balance themselves. The paper is well-structured but the grammar and writing need further improvement. Also, there are a few concerns that need to be addressed to strengthen this work.
Major concern:
1. I understand the purpose of the proposed method is to visualize the point of contact and body supporting force during the daily environment. I am unsure about how to use this to design a better environment. For example, with this method, the user only have limited options on what they can rely on to balance themselves - the environment is fixed. Therefore, we only know for example, a shelf is better than handrail in this household. But maybe the height of the handrail is the problem that prevents the person from using it. We don't know. So I am curious to see how this can turn into a technology that can really help with the decision choice.
2. The FingerTPS measuring range is 4.55-22.73 kg. What does the data look like when the force is out of the range? Also, is there more analysis on how the supporting force look like (e.g., magnitude) for different cases? Related to this question, Figure 14 shows a comparison force plot, The peak force is below 5 N which is way less than the minimum force range 4.55 kg. Is the data reliable when the force is that low?
3. Why did the authors choose to round the timestamps of all sensors to the nearest 10 ms? Given the difference in sampling rates, wouldn’t repeated rounding introduce cumulative timing errors and potentially degrade temporal fidelity? Wouldn’t resampling all data to a fixed rate (e.g., 100 Hz) using interpolation or filtering be a more robust approach to align the data without such abrupt shifts?
4. Can the authors elaborate on how do they choose the region to place the force sensor on? Why there is no sensor near the hypothenar region or the little finger?
Minor concerns:
1. The reviewer recommends a clearer version of the experiment setup to demonstrate the all the measuring sensors/devices. For example, where do the RGB-D camera and Ipad located. Fig. 2 and 3 do not provide a sufficient overview to understand the setup easily.
2. This manuscript consists some grammar problems. For example, on page 4 Section 2.1, "...including not only the palm, but also the fingers only". The reviewer recommend another round of thorough review to improve the clarity of the text.
3. The reviewer recommends to list the math equations or a pseudo code to describe the last paragraph of 2.2.1 for the function to align the coordinate system. It's not clear by just reading the text.
Author Response
Major concern:
Comment1: I understand the purpose of the proposed method is to visualize the point of contact and body supporting force during the daily environment. I am unsure about how to use this to design a better environment. For example, with this method, the user only have limited options on what they can rely on to balance themselves - the environment is fixed. Therefore, we only know for example, a shelf is better than handrail in this household. But maybe the height of the handrail is the problem that prevents the person from using it. We don't know. So I am curious to see how this can turn into a technology that can really help with the decision choice.
Response1:
Thank you for your valuable comment. We agree that the current environment is fixed in each household, and the system as implemented does not directly manipulate environmental variables such as height or layout. However, as described in the revised Discussion section, we clarified that the proposed system allows us to observe actual user interactions with both assistive and non-assistive objects in their natural contexts. Through accumulating this kind of behavioral data, we aim to identify patterns of embodied strategies and mismatches between intended design and actual usage (see Section 5.1).
Moreover, we added that this system can support retrospective evaluation of existing support structures (e.g., handrail placement and height) and predictive modeling in future iterations, as more data are accumulated. We believe that these steps contribute to practical design recommendations and the development of intuitive, user-centered support tools.
Comment2: The FingerTPS measuring range is 4.55-22.73 kg. What does the data look like when the force is out of the range? Also, is there more analysis on how the supporting force look like (e.g., magnitude) for different cases? Related to this question, Figure 14 shows a comparison force plot, The peak force is below 5 N which is way less than the minimum force range 4.55 kg. Is the data reliable when the force is that low?
Response2: We appreciate your careful observation regarding the sensor range and force magnitude. As you noted, the manufacturer of the FingerTPS sensor defines the nominal range as 4.55–22.73 kg (approximately 44.6–223 N). However, our system applies a custom calibration process using low-range loads and regression fitting, enabling us to derive accurate force values in the low-Newton range. This calibration process is now described in Section 2.3.
Comment3: Why did the authors choose to round the timestamps of all sensors to the nearest 10 ms? Given the difference in sampling rates, wouldn’t repeated rounding introduce cumulative timing errors and potentially degrade temporal fidelity? Wouldn’t resampling all data to a fixed rate (e.g., 100 Hz) using interpolation or filtering be a more robust approach to align the data without such abrupt shifts?
Response3: Thank you for your insightful feedback. We have revised the manuscript to clarify our rationale for rounding all timestamps to the nearest 10 milliseconds, including a discussion of the sampling rates, the independent operation of sensors, and the practical trade-offs involved.
Comment4: Can the authors elaborate on how do they choose the region to place the force sensor on? Why there is no sensor near the hypothenar region or the little finger?
Response4: Thank you for your insightful comment. We have added an explanation in Section 2.2.1 to clarify our rationale for sensor placement. Due to the hardware limitation of six force sensor units per hand, we prioritized locations based on preliminary trials in which the first author performed typical body-supporting behaviors (e.g., leaning, gripping furniture). From these observations, the thenar region, central palm, and finger pads from the index to ring fingers were consistently involved in contact and pressure. As a result, these regions were selected for sensor placement, while the hypothenar region and little finger were excluded despite their occasional involvement. We acknowledge this as a limitation and will explore expanding sensor coverage in future work.
Minor concern:
Comment5: The reviewer recommends a clearer version of the experiment setup to demonstrate the all the measuring sensors/devices. For example, where do the RGB-D camera and Ipad located. Fig. 2 and 3 do not provide a sufficient overview to understand the setup easily.
Response5: Thank you for your suggestion. In response, we have revised the figures related to the experimental setup. A new schematic diagram (Figure 2) has been added to clearly illustrate the location and role of each sensor and device, including the RGB-D camera and iPad. This updated figure now provides a comprehensive overview of the measurement environment, helping readers better understand the setup.
Comment6: This manuscript consists some grammar problems. For example, on page 4 Section 2.1, "...including not only the palm, but also the fingers only". The reviewer recommend another round of thorough review to improve the clarity of the text.
Response6: Thank you for pointing this out. We performed a thorough proofreading of the manuscript and revised the sentence you mentioned (Section 2.1) along with many other instances of unclear or awkward phrasing. A professional grammar and language review has also been conducted to improve the overall clarity and readability of the text.
Comment7: The reviewer recommends to list the math equations or a pseudo code to describe the last paragraph of 2.2.1 for the function to align the coordinate system. It's not clear by just reading the text.
Response7: Thank you for your suggestion. We have substantially revised the section and reformulated the description based on mathematical expressions to clarify the coordinate alignment procedure.

Reviewer 2 Report
Comments and Suggestions for Authors
This study aims to assess portable technology to measure and visualise body-supporting force vector fields in the daily environment. While the topic is relevant and the approach rather innovative, the study needs better structuring, clarity, and quantitative reporting. This manuscript has the potential to make an impactful contribution to Sensors and the field of ageing, design, and wearable technology, if the authors apply suggested improvements:
Abstract:
- The term “body-supporting force vector field” may be unclear to general readers without a biomechanics background. Try to simplify this.
- The main outcome metrics (e.g., positional error, force magnitudes) are not explicitly mentioned.
- The system's significance (i.e., how it advances current methods) is stated in general terms, not quantitatively. Please add numerical highlights from results (e.g., error rates, number of subjects) to increase the abstract's readability.
Introduction:
- Overall, the introduction is long, hard to read, and not well-structured. It would benefit from more concise paragraphing. Shorten the introduction to about 1.5 pages. Reduce the public health background and focus more on the limitations of embedded sensors vs. wearable tech.
- The research gap in current methods related to this study is repeated across multiple paragraphs. Streamlining is needed to focus on the research gap where it is required.
- State a straightforward research question and hypothesis by the end of the introduction.
Methods:
- Sample size is relatively low. Please describe how you got this number of participants.
- Some terms (e.g., “gravity-horizontal component ratio,” “coordinate alignment,” etc.) are not clearly defined until deep into the technical section. The authors could provide a summary diagram or table of the data acquisition workflow (sensor fusion, alignment, visualisation).
- There is no description of how behaviours were standardised across participants—how was variability in the actions handled? Please add more details on software processing: data cleaning, outlier handling, and segmentation of force events.
- No formal performance metrics (e.g., RMSE of fusion, test-retest reliability) are reported. Please include reliability/validity metrics or cite supporting literature for sensor accuracy beyond anecdotal error estimations.
Results/Discussion:
- It is not adequate for a journal of this rank to merge results and discussion. Please separate these sections and focus more on the discussion part. There is virtually no discussion at all—mostly results and speculation. Please use adequate literature to make this manuscript more scientific and less analytical.
- There is no quantitative comparison to a gold standard, such as pressure mats or load cells.
- The force magnitudes are small (~4 N) and might be influenced by noise or sensitivity limits. This issue is not addressed.
- Some figures are too small or cluttered. It is hard to interpret vector directions or force magnitudes. Also, legends and axis labels are occasionally missing or too vague.
- Based on the improvements suggested above, restructure and rewrite limitations and conclusions.
Author Response
We sincerely appreciate the reviewers’ thoughtful comments and constructive suggestions, which have greatly improved the clarity and rigor of our manuscript. Below, we provide point-by-point responses to each comment.
Comment1: The term “body-supporting force vector field” may be unclear to general readers without a biomechanics background. Try to simplify this.
Response1: Thank you for your valuable feedback. In response, we have revised the abstract to replace the term “body-supporting force vector field” with a more intuitive and accessible description:
“...how the body interacts with real-world objects for support...”
This phrasing avoids specialized biomechanical jargon while preserving the original meaning. We believe this change enhances readability and accessibility, especially for interdisciplinary readers who may not be familiar with biomechanical terminology.
Comment2: The main outcome metrics (e.g., positional error, force magnitudes) are not explicitly mentioned.
Response2: We appreciate your observation. As this study primarily focuses on developing a novel system for in-situ, real-environment behavioral visualization, our key outcomes are behavioral distinctions and spatial support mapping rather than clinical or biomechanical accuracy metrics.
To reflect this emphasis, we revised the abstract as follows:
“...successfully distinguished support interactions with specific furniture (e.g., doorframes, shelves) ...”
This highlights the system’s practical performance in detecting context-specific support behaviors without overstating precision-based metrics. More technical evaluations, including alignment accuracy, are detailed in the main text.
Comment3: The system's significance (i.e., how it advances current methods) is stated in general terms, not quantitatively. Please add numerical highlights from results (e.g., error rates, number of subjects) to increase the abstract's readability.
Response3: Thank you for your comment. We agree that expressing the system’s advancement in quantitative terms can improve clarity. However, the primary contribution of this study lies not in surpassing existing tools on predefined metrics, but in enabling previously unattainable measurements—namely, the 3D visualization of body-support interactions in real residential environments without modifying the space.
To reflect this, we revised the abstract to emphasize the system’s ability to distinguish support interactions with specific furniture (e.g., doorframes, shelves), and noted the use of 13 participants across 9 households. These additions convey the practical scope and granularity of the system.
While we do not provide conventional performance metrics (e.g., error rates), we believe the ability to capture meaningful physical interaction data in uncontrolled, real-life environments represents a substantial advancement over prior methods.
Comment4: Overall, the introduction is long, hard to read, and not well-structured. It would benefit from more concise paragraphing. Shorten the introduction to about 1.5 pages. Reduce the public health background and focus more on the limitations of embedded sensors vs. wearable tech.
Response4: Thank you for this constructive feedback. We have significantly revised the Introduction to improve its readability and structure. Specifically, we reduced the length to approximately 1.5 pages, eliminated redundant explanations, and streamlined the paragraphing for clarity. Additionally, we minimized the public health background and placed greater emphasis on the limitations of embedded sensing systems in contrast to wearable technologies. We believe these changes now provide a more focused and accessible introduction for readers.
Comment5: The research gap in current methods related to this study is repeated across multiple paragraphs. Streamlining is needed to focus on the research gap where it is required.
Response5: We appreciate your observation regarding redundancy in presenting the research gap. In the revised manuscript, we consolidated the discussion of related work and clarified the specific research gap our study addresses. This streamlined approach reduces repetition and helps the reader understand the novelty and necessity of our proposed method in a more direct manner.
Comment6: State a straightforward research question and hypothesis by the end of the introduction.
Response6: Thank you for pointing this out. We have revised the concluding paragraph of the Introduction to clearly state the research question and hypothesis. Specifically, we now highlight that the research question centers on how body-supporting behaviors in natural daily environments can be effectively visualized using a portable multimodal sensing system. The hypothesis is that such visualization enables the identification of meaningful contact regions and force patterns, which could inform future environmental design interventions.
Comment7: Sample size is relatively low. Please describe how you got this number of participants.
Response7: Thank you for your comment. We acknowledge that the sample size of 13 participants is relatively small. As described in the revised manuscript (Section 5.3, Limitations). Participants were selected from 9 different households to reflect variability in environmental settings. While the study provides a proof-of-concept demonstration of the system’s feasibility and utility, we agree that future work should involve a larger cohort to allow for statistical generalization and broader applicability.
Comment8: Some terms (e.g., “gravity-horizontal component ratio,” “coordinate alignment,” etc.) are not clearly defined until deep into the technical section. The authors could provide a summary diagram or table of the data acquisition workflow (sensor fusion, alignment, visualization).
Response8: We appreciate this valuable comment. To improve clarity, we have added a summary diagram that outlines the entire data processing workflow, including sensor fusion, coordinate alignment, and visualization steps (see revised Figure 2).
Comment9: There is no description of how behaviors were standardized across participants—how was variability in the actions handled? Please add more details on software processing: data cleaning, outlier handling, and segmentation of force events.
Response9: Thank you for your valuable suggestion. We have added a more detailed explanation of the data processing procedures in Section 3.2 of the revised manuscript. Since the study aimed to capture natural, everyday behaviors, we did not instruct participants to perform specific standardized tasks. Instead, we recorded a variety of daily actions in each participant's home and subsequently classified and analyzed the data based on behavior categories.
We also added information on how noise and outliers were handled: specifically, we removed data segments with missing sensor signals or unstable posture estimation. Force events were segmented based on thresholds in force signal magnitude and temporal continuity. These updates are now reflected in the revised manuscript.
Comment10: No formal performance metrics (e.g., RMSE of fusion, test-retest reliability) are reported. Please include reliability/validity metrics or cite supporting literature for sensor accuracy beyond anecdotal error estimations.
Response10: Thank you for pointing this out. While the primary aim of this study was not to evaluate sensor performance, we acknowledge the importance of clarifying the accuracy and reliability of the data. In the revised manuscript (Section 2.2), we have added a description of the known accuracy specifications for the depth sensor and force sensor, as provided by the manufacturers and supported by prior validation studies. For example, the depth sensor used has a positional error of less than 20 mm within the range of use, and the FingerTPS force sensor has demonstrated sufficient resolution for detecting low-magnitude forces (~4 N), as also shown in our experimental results. These references provide assurance of the validity of the measurements used in this study.
Comment11: It is not adequate for a journal of this rank to merge results and discussion. Please separate these sections and focus more on the discussion part. There is virtually no discussion at all—mostly results and speculation. Please use adequate literature to make this manuscript more scientific and less analytical.
Response11: Thank you for this important comment. In the revised manuscript, we have separated the “Results” and “Discussion” sections into distinct sections (Sections 3 and 4, respectively). The revised Results section now focuses solely on the objective presentation of findings, without interpretation. In the new Discussion section, we interpret these findings considering relevant literature, theoretical context, and design implications. We have cited additional references to support our interpretations and explicitly discuss how our observations relate to prior studies on fall prevention and environmental interaction. These changes have improved the clarity, scientific rigor, and structure of the manuscript.
Comment12: There is no quantitative comparison to a gold standard, such as pressure mats or load cells.
Response12: Thank you for your valuable comment. While the current study did not conduct a new quantitative comparison with standard measurement tools such as pressure mats or load cells, we have clarified this point in the revised manuscript. Specifically, we added a reference to our previous study (Nomura et al., Sensors 2022), in which the same visualization system was evaluated using a pressure mat. In that study, the average localization error between the estimated contact area and the pressure mat output was 27 mm (SD: 21 mm), with a maximum error of approximately 50 mm. We believe that this level of spatial accuracy is sufficient for distinguishing coarse contact locations (e.g., center vs. edge of a desk) in body-supporting scenarios. This explanation has been added to the Method section to address your concern.
Comment13: The force magnitudes are small (~4 N) and might be influenced by noise or sensitivity limits. This issue is not addressed.
Response13: We appreciate your careful observation regarding the sensor range and force magnitude. As you noted, the manufacturer of the FingerTPS sensor defines the nominal range as 4.55–22.73 kg (approximately 44.6–223 N). However, our system applies a custom calibration process using low-range loads and regression fitting, enabling us to derive accurate force values in the low-Newton range. This calibration process is now described in Section 2.3 (p.6).
Comment14: Some figures are too small or cluttered. It is hard to interpret vector directions or force magnitudes. Also, legends and axis labels are occasionally missing or too vague.
Response14: Thank you for pointing this out. We have revised the visualization figures to enhance readability. Specifically, we enlarged the figures (e.g., Figures 9–13), adjusted the density of overlaid vectors to reduce visual clutter, and clarified color legends and axis labels. Where necessary, we also added unit information and explanatory annotations to better communicate vector direction and magnitude.
Comment15: Based on the improvements suggested above, restructure and rewrite limitations and conclusions.
Response15: Thank you for your suggestion. We have revised and restructured the Limitations and Conclusions sections to better reflect the clarified contributions, data processing procedures, and implications for future research. The updated sections now emphasize the novelty of capturing real-world support behaviors, acknowledge sample size and sensor constraints, and outline directions for design application and data-driven modeling. Please see Sections 5.1 and 5.2 in the revised manuscript for the updated content.
Round 2
Reviewer 1 Report
Comments and Suggestions for Authors
Minor suggestions:
- Please make sure the spacing are correct across the manuscript. For example, ‘100 Hz’ vs. ‘100Hz’.
- The figures look really blurred after zooming in, especially those with images such as Figure 2. Please consider to use the vector graphs and improve the image quality.
Author Response
We sincerely appreciate your careful reading of our manuscript and your constructive feedback. Your comments have been extremely helpful in improving the clarity, accuracy, and overall quality of our work. Thank you again for your time and thoughtful suggestions.
Comment1: Please make sure the spacing are correct across the manuscript. For example, ‘100 Hz’ vs. ‘100Hz’.
Response1: Thank you for pointing this out. We carefully reviewed the entire manuscript and corrected spacing issues to ensure consistency. For instance, instances such as “100Hz” have been revised to “100 Hz” throughout the text.
Comment2: The figures look really blurred after zooming in, especially those with images such as Figure 2. Please consider to use the vector graphs and improve the image quality.
Response2: Thank you for your valuable feedback. We have replaced the blurred figures, including Figure 2, with higher-resolution or vector-based versions to enhance clarity and readability when zoomed in.
Reviewer 2 Report
Comments and Suggestions for Authors
I’m pleased with the authors' responses and corrections they made to improve this manuscript.
Author Response
Thank you very much for your positive feedback. We are pleased to hear that the revisions and responses have met your expectations. We sincerely appreciate your valuable comments and suggestions, which helped us improve the clarity and quality of the manuscript.